

# Tracing hepatitis B virus (HBV) genotype B5 (formerly B6) evolutionary history in the circumpolar Arctic through phylogeographic modelling

Remco Bouckaert[1], Brenna C. Simons[2], Henrik Krarup[3], T. Max Friesen[4] and Carla Osiowy[5,6]

[1] Department of Computer Science, University of Auckland, Auckland, New Zealand
[2] Alaska Native Tribal Health Consortium, Anchorage, AK, United States of America
[3] Section of Molecular Diagnostics, Clinical Biochemistry, Aalborg University Hospital, Aalborg, Denmark
[4] Department of Anthropology, University of Toronto, Toronto, Ontario, Canada
[5] National Microbiology Laboratory, Public Health Agency of Canada, Winnipeg, Manitoba, Canada
[6] Department of Medical Microbiology and Infectious Diseases, University of Manitoba, Winnipeg, Manitoba, Canada

Corresponding author
Carla Osiowy,
carla.osiowy@phac-aspc.gc.ca

## ABSTRACT

**Background**. Indigenous populations of the circumpolar Arctic are considered to be endemically infected ($>$2% prevalence) with hepatitis B virus (HBV), with subgenotype B5 (formerly B6) unique to these populations. The distinctive properties of HBV/B5, including high nucleotide diversity yet no significant liver disease, suggest virus adaptation through long-term host-pathogen association.

**Methods**. To investigate the origin and evolutionary spread of HBV/B5 into the circumpolar Arctic, fifty-seven partial and full genome sequences from Alaska, Canada and Greenland, having known location and sampling dates spanning 40 years, were phylogeographically investigated by Bayesian analysis (BEAST 2) using a reversible-jump-based substitution model and a clock rate estimated at $4.1 \times 10^{-5}$ substitutions/site/year.

**Results**. Following an initial divergence from an Asian viral ancestor approximately 1954 years before present (YBP; 95% highest probability density interval [1188, 2901]), HBV/B5 coalescence occurred almost 1000 years later. Surprisingly, the HBV/B5 ancestor appears to locate first to Greenland in a rapid coastal route progression based on the landscape aware geographic model, with subsequent B5 evolution and spread westward. Bayesian skyline plot analysis demonstrated an HBV/B5 population expansion occurring approximately 400 YBP, coinciding with the disruption of the Neo-Eskimo Thule culture into more heterogeneous and regionally distinct Inuit populations throughout the North American Arctic.

**Discussion**. HBV/B5 origin and spread appears to occur coincident with the movement of Neo-Eskimo (Inuit) populations within the past 1000 years, further supporting the hypothesis of HBV/host co-expansion, and illustrating the concept of host-pathogen adaptation and balance.

## INTRODUCTION

Endemic infection (>2%) with hepatitis B virus (HBV) has been historically observed throughout Inuit and Alaska Native populations of the western circumpolar Arctic (*McMahon, 2004*; *Minuk & Uhanova, 2003*; *Tulisov et al., 2007*), although ongoing vaccination programs, starting in the mid-1980's to mid-1990's, have or are expected to reduce prevalence to non-endemic levels (*Huynh et al., 2014*; *McMahon et al., 2011*). Studies have shown several HBV genotypes circulating within Inuit or Alaska Native people of the circumpolar Arctic; however, subgenotype B5 (HBV/B5; formerly B6 (*Kramvis, 2014*)) is unique to this population and has not been found elsewhere (*Osiowy, Simons & Rempel, 2013*). While chronic HBV infection often results in liver cirrhosis or hepatocellular carcinoma, HBV/B5 chronic infection is infrequently associated with serious adverse effects (*Krarup et al., 2008*; *Minuk et al., 2013*; *Sakamoto et al., 2007*) and rather, results in a 'benign' outcome. This clinical association, together with the distinct nucleotide diversity and mutation rate observed with HBV/B5 compared to other HBV genotypes infecting Inuit and Alaska Native populations (*Kowalec et al., 2013*), suggests potential pathogen attenuation within the population due to host-pathogen co-evolution (*Paraskevis et al., 2013*; *Tedder et al., 2013*). There is support for human/HBV coevolution (*Paraskevis et al., 2013*; *Suh et al., 2013*), particularly with respect to the distinct geographic distribution of genotypes observed throughout the world and association with specific ethnic groups. In particular HBV genotypes associated with remote or isolated indigenous populations provide a more stable foundation to investigate the coevolving relationship based on the historical HBV endemicity within these populations and their status as the first peoples of a geographic region (*Littlejohn, Locarnini & Yuen, 2016*; *Zehender et al., 2014*). In order to investigate the origin, evolution and spread of HBV/B5, spatial and temporal phylogenetic analysis of HBV sequences obtained throughout the circumpolar Arctic was performed. This analysis should further our understanding of the natural history, origin and evolutionary rate of HBV/B5, which in turn provides increased understanding of HBV evolutionary history and the concept of host-pathogen balance through co-evolution. Through incorporation of viral population size and evolutionary rate estimates, the virus strain could be traced through the circumpolar region to infer the time to the most recent common ancestor (tMRCA) and putative dispersal over time to support long-term association of HBV/B5 with populations indigenous to the western circumpolar region.

## MATERIALS AND METHODS

### Serum samples, HBV DNA extraction and sequencing analysis

Twenty-two serum samples from various locations in Nunavut were collected from HBV/B5-infected individuals having self-identified as Inuit. All specimens were described previously in studies for which informed patient consent was obtained for HBV molecular analysis and study approval was granted from institutional ethics review boards (Health Canada/Public Health Agency of Canada research ethics board protocol number REB-2006-0048 and REB-2012-0062; University of Manitoba research ethics board approval number HS15821; *Huynh et al., 2014*; *Larke et al., 1987*; *Minuk et al., 2013*; *Osiowy, Larke*

& *Giles, 2011*). HBV DNA was extracted from 200 µl sera by SDS-proteinase K lysis and phenol chloroform extraction methods (*Osiowy, 2002*) and resuspended in 30 µl sterile, nuclease-free water. Amplification of the full HBV genome was performed using a high fidelity polymerase (Roche Expand High Fidelity[Plus] System, Roche Diagnostics, Laval, QC, Canada) as described previously (*Osiowy et al., 2010*). Samples having an HBV viral load precluding full genome amplification and sequencing were partially sequenced using primers and methods described previously (*Minuk et al., 2012*). Specific amplicons were gel-purified prior to cycle sequencing with an AB 3730 XL DNA Analyzer using Big Dye 3.1 terminator chemistry (Thermo Fisher Scientific, Burlington, ON, Canada). Sequences were assembled and analysed using DNA sequence analysis software (Lasergene software suite v 10.0, DNASTAR, Madison, WI, USA). Sequences were submitted to GenBank under accession numbers KP659234–KP659255. Nine other sequences previously described from individuals residing in Nunavut were included (JN792894, JN792896, JN792897, JN792900, JN792901, DQ463795, DQ463796, DQ463799, DQ463802; *Kowalec et al., 2013*; *Osiowy et al., 2006*). Thus, a total of 31 partial and full genome HBV sequences from Nunavut were included in the analysis (Table 1).

Ten serum samples from various locations in Alaska were collected and the HBV DNA extracted and sequenced as described for Nunavut specimens. Participants were enrolled statewide by the Liver Disease and Hepatitis Program at the Alaska Native Tribal Health Consortium (Anchorage, AK). This study was approved by the Alaska Area (Indian Health Service) and the Centers for Disease Control and Prevention Institutional Review Boards (Protocol number AAIRB 2001-07-022/1996-01-001). The study and manuscript were approved by the Alaska Native Tribal Health Consortium and the Southcentral Foundation Board of Directors. All participants provided written informed consent. Sequences were submitted to GenBank under accession numbers KP659219–KP659228. Six other sequences previously described from individuals residing in Alaska were included (AB287314–AB297319; *Sakamoto et al., 2007*). Thus a total of 16 partial and full genome HBV sequences from Alaska were included in the analysis (Table 1).

Four serum specimens collected from East Greenland were extracted for HBV DNA and sequenced as described for Nunavut specimens. These specimens were collected previously for the study by *Krarup et al. (2008)*, in which ethics approval was granted by the Commission for Scientific Research in Greenland (Approval number 505–99), and written informed consent was obtained from each participant. Sequences were submitted to GenBank under accession numbers KP659230–KP659233. Six other sequences previously described from individuals located in West Greenland were included (AB287320–AB287325; *Sakamoto et al., 2007*). Thus a total of 10 partial and full genome HBV sequences from Greenland were included in the analysis (Table 1).

Methods for specimen and data collection and analysis were carried out in accordance with the Tri-Council Policy on Ethical Conduct for Research Involving Humans (Canadian Institutes of Health Research, Natural Sciences and Engineering Research Council of Canada, and the Social Sciences and Humanities Research Council of Canada).
**Table 1** Sample date, location, and molecular properties of sequences included in the analysis.

| GenBank accession # | HBV sub-genotype | G1896A | Full or partial genome | Partial genome region included (nucleotide)[a] | Location information | Sample dates | Reference |
|---|---|---|---|---|---|---|---|
| **ALASKA** | | | | | | | |
| KP659219 | B5 | A1896 | Full | | Alaska | 07-Feb-2006 | Present study |
| KP659220 | B5 | A1896 | Full | | Alaska | 26-Oct-2005 | Present study |
| KP659221 | B5 | A1896 | Full | | Alaska | 25-Jan-2005 | Present study |
| KP659222 | B5 | A1896 | Partial | 153–881; 1839–3215 | Alaska | 12-May-2003 | Present study |
| KP659223 | B5 | A1896 | Full | | Alaska | 15-Sep-2003 | Present study |
| KP659224 | B5 | A1896 | Full | | Alaska | 10-Dec-2008 | Present study |
| KP659225 | B5 | G1896 | Partial | 13–866; 1360–1774; 2347–3142 | Alaska | 01-Sep-2008 | Present study |
| KP659226 | B5 | G1896 | Partial | 13–884; 1002–1397; 2338–3168 | Alaska | 18-Jun-2003 | Present study |
| KP659227 | B5 | G1896 | Partial | 155–834 | Alaska | 14-Dec-2006 | Present study |
| KP659228 | B5 | G1896 | Partial | 155–834 | Alaska | 29-Apr-2003 | Present study |
| AB287314 | B5 | G1896 | Full | | Alaska | 23-Oct-1973 | *Sakamoto et al. (2007)* |
| AB287315 | B5 | G1896 | Full | | Alaska | 19-Mar-1975 | *Sakamoto et al. (2007)* |
| AB287316 | B5 | A1896 | Full | | Alaska | 23-Oct-1973 | *Sakamoto et al. (2007)* |
| AB287317 | B5 | A1896 | Full | | Alaska | 21-Mar-1974 | *Sakamoto et al. (2007)* |
| AB287318 | B5 | A1896 | Full | | Alaska | 01-Jan-2004 | *Sakamoto et al. (2007)* |
| AB287319 | B5 | A1896 | Full | | Alaska | 29-Jan-2004 | *Sakamoto et al. (2007)* |
| **CANADA** | | | | | | | |
| KP659234 | B5 | G1896 | Full | | West Nunavut | 08-Oct-1983 | *Osiowy, Larke & Giles (2011)*; Present study |
| KP659235 | B5 | G1896 | Full | | West Nunavut | 08-Oct-1983 | *Osiowy, Larke & Giles (2011)*; Present study |
| KP659236 | B5 | A1896 | Partial | 1–258; 803–1775; 1838–3215 | West Nunavut | 21-Apr-1984 | *Osiowy, Larke & Giles (2011)*; Present study |
| KP659237 | B5 | G1896 | Full | | East Nunavut | 23-Oct-1983 | *Osiowy, Larke & Giles (2011)*; Present study |
| KP659238 | B5 | A1896 | Partial | 1–1770; 2375–3215 | West Nunavut | 06-Mar-1984 | *Osiowy, Larke & Giles (2011)*; Present study |

| GenBank accession # | HBV sub-genotype | G1896A | Full or partial genome | Partial genome region included (nucleotide)[a] | Location information | Sample dates | Reference |
|---|---|---|---|---|---|---|---|
| KP659239 | B5 | A1896 | Partial | 1–1778; 1839–3215 | West Nunavut | 16-Mar-1984 | *Osiowy, Larke & Giles (2011)*; Present study |
| KP659240 | B5 | A1896 | Partial | 1–1793; 1838–3215 | East Nunavut | 01-May-2012 | *Minuk et al. (2013)*; Present study |
| KP659241 | B5 | A1896 | Partial | 1–254; 803–1792; 1833–3215 | East Nunavut | 28-Apr-2012 | *Minuk et al. (2013)*; Present study |
| KP659242 | B5 | A1896 | Partial | 1–253; 780–1770; 1840–2848; 2867–3215 | East Nunavut | 27-Apr-2012 | *Minuk et al. (2013)*; Present study |
| KP659243 | B5 | A1896 | Partial | 1–898; 969–1333; 1666–2405; 2825–3215 | East Nunavut | 15-Sep-1983 | *Osiowy, Larke & Giles (2011)*; Present study |
| KP659244 | B5 | A1896 | Partial | 1–1792; 1839–3215 | East Nunavut | 26-Apr-1983 | *Osiowy, Larke & Giles (2011)*; Present study |
| KP659245 | B5 | A1896 | Full | | East Nunavut | 09-May-2012 | *Minuk et al. (2013)*; Present study |
| KP659246 | B5 | A1896 | Full | | East Nunavut | 23-Oct-1983 | *Osiowy, Larke & Giles (2011)*; Present study |
| KP659247 | B5 | G1896 | Full | | East Nunavut | 05-May-1983 | *Osiowy, Larke & Giles (2011)*; Present study |
| KP659248 | B5 | A1896 | Full | | East Nunavut | 29-Apr-2013 | *Huynh et al. (2014)*; Present study |
| KP659249 | B5 | A1896 | Full | | East Nunavut | 01-May-2013 | *Huynh et al. (2014)*; Present study |
| KP659250 | B5 | A1896 | Full | | East Nunavut | 17-Jun-2013 | *Huynh et al. (2014)*; Present study |
| KP659251 | B5 | A1896 | Full | | East Nunavut | 10-Apr-2013 | *Huynh et al. (2014)*; Present study |
| KP659252 | B5 | A1896 | Full | | East Nunavut | 22-May-2013 | *Huynh et al. (2014)*; Present study |
| KP659253 | B5 | A1896 | Full | | East Nunavut | 22-May-2013 | *Huynh et al. (2014)*; Present study |

| GenBank accession # | HBV sub-genotype | G1896A | Full or partial genome | Partial genome region included (nucleotide)[a] | Location information | Sample dates | Reference |
|---|---|---|---|---|---|---|---|
| KP659254 | B5 | A1896 | Full | | East Nunavut | 27-May-2013 | *Huynh et al. (2014)*; Present study |
| KP659255 | B5 | A1896 | Full | | West Nunavut | 17-Jun-2013 | *Huynh et al. (2014)*; Present study |
| JN792894 | B5 | A1896 | Full | | West Nunavut | 01-Sep-2009 | *Kowalec et al. (2013)* |
| JN792896 | B5 | A1896 | Full | | West Nunavut | 01-Sep-2009 | *Kowalec et al. (2013)* |
| JN792897 | B5 | A1896 | Full | | West Nunavut | 01-Sep-2009 | *Kowalec et al. (2013)* |
| JN792900 | B5 | A1896 | Full | | West Nunavut | 01-Sep-2009 | *Kowalec et al. (2013)* |
| JN792901 | B5 | A1896 | Full | | West Nunavut | 01-Sep-2009 | *Kowalec et al. (2013)* |
| DQ463795 | B5 | A1896 | Full | | West Nunavut | 05-Apr-2004 | *Osiowy et al. (2006)* |
| DQ463796 | B5 | A1896 | Full | | West Nunavut | 05-Apr-2004 | *Osiowy et al. (2006)* |
| DQ463799 | B5 | A1896 | Full | | West Nunavut | 05-Apr-2004 | *Osiowy et al. (2006)* |
| DQ463802 | B5 | A1896 | Full | | West Nunavut | 05-Apr-2004 | *Osiowy et al. (2006)* |
| **GREENLAND** | | | | | | | |
| KP659230 | B5 | A1896 | Partial | 1–20; 211–881; 1618–3215 | East Greenland | 01-Nov-1998 | Present study |
| KP659231 | B5 | A1896 | Partial | 159–881; 1362–3184 | East Greenland | 01-Nov-1998 | Present study |
| KP659232 | B5 | A1896 | Partial | 158–880; 1362–3183 | East Greenland | 01-Nov-1998 | Present study |
| KP659233 | B5 | A1896 | Partial | 159–880; 1628–3183 | East Greenland | 01-Nov-1998 | Present study |
| AB287320 | B5 | A1896 | Full | | West Greenland | 01-Jan-1998 | *Sakamoto et al. (2007)* |
| AB287321 | B5 | A1896 | Full | | West Greenland | 01-Jan-1998 | *Sakamoto et al. (2007)* |
| AB287322 | B5 | A1896 | Full | | West Greenland | 01-Aug-2004 | *Sakamoto et al. (2007)* |
| AB287323 | B5 | A1896 | Full | | West Greenland | 01-Jan-1998 | *Sakamoto et al. (2007)* |
| AB287324 | B5 | A1896 | Full | | West Greenland | 01-Aug-2004 | *Sakamoto et al. (2007)* |
| AB287325 | B5 | A1896 | Full | | West Greenland | 01-Aug-2004 | *Sakamoto et al. (2007)* |

**Table 1** (*continued*)

| GenBank accession # | HBV sub-genotype | G1896A | Full or partial genome | Partial genome region included (nucleotide)[a] | Location information | Sample dates | Reference |
|---|---|---|---|---|---|---|---|
| **ASIA** | | | | | | | |
| AB010289 | B1 | G1896 | Full | | Japan | 01-Jan-1993 | *Koseki et al. (1999)* |
| AB010290 | B1 | A1896 | Full | | Japan | 01-Jan-1993 | *Koseki et al. (1999)* |
| AB010291 | B1 | A1896 | Full | | Japan | 01-Jan-1993 | *Koseki et al. (1999)* |
| AB010292 | B1 | A1896 | Full | | Japan | 01-Jan-1993 | *Koseki et al. (1999)* |
| AB073838 | B1 | A1896 | Full | | Japan | 01-Jan-2001 | *Sugauchi et al. (2002)* |
| D23677 | B1 | G1896 | Full | | Japan | 01-Jan-2000 | *Horikita et al. (1994)* |
| D23678 | B1 | A1896 | Full | | Japan | 01-Jan-2000 | *Horikita et al. (1994)* |
| D23679 | B1 | A1896 | Full | | Japan | 01-Jan-2000 | *Horikita et al. (1994)* |
| AB287326 | B1 | A1896 | Full | | Japan | 01-Jan-2006 | *Sakamoto et al. (2007)* |
| AB287327 | B1 | A1896 | Full | | Japan | 01-Jan-2006 | *Sakamoto et al. (2007)* |
| AB602818 | B1 | G1896 | Full | | Japan | 01-Aug-2005 | *Inoue et al. (2011)* |
| FJ386584 | B2 | G1896 | Full | | China | 21-Feb-2008 | *Xu et al. (2011)* |
| FJ386600 | B2 | G1896 | Full | | China | 03-Mar-2008 | *Xu et al. (2011)* |
| FJ386636 | B2 | G1896 | Full | | China | 22-Feb-2008 | *Xu et al. (2011)* |
| GQ924653 | B2 | G1896 | Full | | Malaysia | 27-Jan-2007 | *Meldal et al. (2011)* |

**Notes.**
[a]Nucleotide numbering based on GenBank accession no. DQ463795 (3,215 nt in length).

## Phylogenetic analysis and evolutionary dynamics

All sequences listed in Table 1, including partial and full genome sequences, were aligned using ClustalX (*Thompson et al., 1997*), resulting in an alignment with 3220 sites. Bayesian analysis was performed using Markov Chain Monte Carlo (MCMC) methods implemented in BEAST 2 (*Bouckaert et al., 2014*) for the phylogeny and estimation of the HBV effective population size. In order to facilitate convergence, the MCMC chains were run sufficiently long; at 40 million generations with sampling every 10,000 steps and the first 10% of samples discarded as burn-in. During analysis, the sequences in the West Greenland clade were restricted to be monophyletic. Without monophyly constraint the analysis did not converge, and from our experience, sequences from the western circumpolar regions of Alaska, Canada and Greenland, tend to cluster into monophyletic clades (*Kowalec et al., 2013*; *Sakamoto et al., 2007*). A coalescent tree prior was chosen due to the intra-species analysis and the assumption that sampling across all clades was consistent and accurate. The XML file including all data for BEAST 2 analysis is available as Data S1.

The aligned sequence data consisting of full length and partial sequence was partitioned into eight parts consisting of sites 1–835, 836–1373, 1374–1620, 1621–1900, 1901–2307, 2308–2450, 2451–2847, and 2848–3220, based on gene boundaries in the genome. Since each of these partitions code for different genes, and some even two genes in different reading frames, these partitions will be governed by different evolutionary mechanisms. Therefore, a separate substitution model was used with each partition having its own relative substitution rate. We used the reversible-jump-based substitution model (*Bouckaert, Alvarado-Mora & Pinho, 2013*) with four gamma categories and invariant sites so that uncertainty in the substitution model choice is integrated out. This model jumps between the models F81 (*Felsenstein, 1981*), HKY85 (*Hasegawa, Kishino & Yano, 1985*), TAN93 (*Tamura & Nei, 1993*), TIM (*Posada, 2003*), EVS (*Drummond & Bouckaert, 2015*) and GTR (*Tavaré, 1986*) and automatically estimates the model parameters during the MCMC.

The phylogenetic tree was prepared using DensiTree implemented in BEAST 2, which has the advantage of being able to visualize the uncertainty in both node heights and topology, such that a qualitative analysis of all tree sets can be made (*Heled & Bouckaert, 2013*). Bayesian skyline plot analysis was used to estimate the relative HBV/B5 population size through time (*Drummond et al., 2005*). A coalescent tree prior is used, which is an appropriate prior for within species samples. Furthermore, it uses a non-parametric population function, so there is no commitment to a parametric population function; that is, a population history, such as an exponentially growing population or constant population, is not assumed beforehand. Default settings for the Bayesian skyline plot (five intervals, default hyper priors) were used.

## Geographic analysis

A joint geographic and phylogenetic analysis of aligned partial and full genome sequences listed in Table 1 using the landscape aware geographic model (*Bouckaert et al., 2012*) was performed using BEAST 2. Essentially, the model uses a random walk based on successive steps in a random direction to model the spread of organisms across a landscape. The area of interest throughout the western circumpolar Arctic region that is being investigated contains many large landmasses among long stretches of coastline. The landscape aware model was configured to distinguish between sea, inland, and coastal regions and assumed fast (10-fold higher) dispersion rates along the coast compared to dispersion rates inland. Furthermore, there is a reluctance to get into water, but once in water, the rate of dispersal is very high (10-fold higher than along the coast).

A prior on the location of the root of the HBV phylogeographic summary tree was used to enforce its location on the Asian side of the Bering Strait, in order to minimize the geographical area used for the landscape aware model, and to parallel the likely HBV/B5 origin and dispersal from Asia through the Bering Strait region (*Paraskevis et al., 2015*); thus, the Asian HBV sequences do not impact the geography of the North American and Greenlandic HBV sequences.

During the MCMC run, locations of internal nodes are sampled. To position a branch onto the map, the sample location information is used for the start and end of a branch. To find the location of the mid-point of the branch of length $t$, the location in a $32 \times 32$ grid

that maximises the probability of going from a start location to that grid point after time $t/2$, multiplied by the probability of going from an end location to that grid point after time $t/2$, is determined. The branch is continually split recursively until all intermediate locations are neighbouring grid points. This determines the most probable path for a branch, which can then be visualised by straight lines connecting grid points. Note that Bayesian analyses do not produce single summary trees (visualised as a blue solid line) but distributions over trees are represented. These are samples from the posterior in the form of a set of trees. Each of these trees follows a somewhat different trajectory and can be visualised individually to get an impression of the uncertainty in the distribution route as well as the area that potentially was visited. Instead of drawing each tree using an opaque line (as was done for the summary tree) a colored dot is drawn using translucency for every grid point in the $32 \times 32$ grid traversed by the trajectory associated with the tree. Time depth can be visualised using colour, ranging from light blue (older) to red (younger).

## RESULTS

### HBV/B5 phylogenetic analysis

A total of 72 HBV sequences were included in the phylogenetic analysis; fifty-seven HBV/B5 sequences, of which 40 were full genome sequence, and 11 HBV/B1 and 4 HBV/B2 GenBank-derived full genome sequences used as an outgroup (Table 1). HBV/B1 and B2 sequences were included in the phylogenetic analysis to delineate the ancestral foundation and putative geographic origin of HBV/B5, as HBV/B1 sequences are most phylogenetically similar to HBV/B5 sequences, while HBV/B2 sequences are more distantly related to both B1 and B5 (*Sakamoto et al., 2007*). Both HBV/B1 and B2 are localized to regions in Asia (*Sakamoto et al., 2007*) but have not been observed to circulate in Arctic regions. Location information, sample dates and the presence of the precore stop codon mutation (G1896A) for all sequences are listed in Table 1, as are the genomic regions covered for partial sequences. Due to privacy concerns given the small population sizes of the Inuit communities, detailed location information is not given for certain samples not previously described.

Figure 1 shows the relative substitution rates and preferred substitution models for the partitioned data following reversible-jump-based substitution model analysis. Since HBV has a complex genome with many overlapping open reading frames, partitioning can be expected to give a more realistic model of substitution, as each subgenomic region has a different substitution rate (*Bouckaert, Alvarado-Mora & Pinho, 2013*). The resulting general trends are similar to earlier findings of *Bouckaert, Alvarado-Mora & Pinho (2013)*, such as the requirement for complex substitution models in sequence areas where genes overlap, and higher rates where there is no overlap.

The uncorrelated log-normal relaxed clock (*Drummond et al., 2006*) fit the data better than the strict clock, as resulting likelihoods and posterior were non-overlapping. Furthermore, the mean coefficient of correlation of 0.63 (95% highest probability density (HPD) Interval [0.52, 0.75]) suggests that the strict clock could be dismissed. The clock was calibrated using archeological evidence from human history, and since humans are

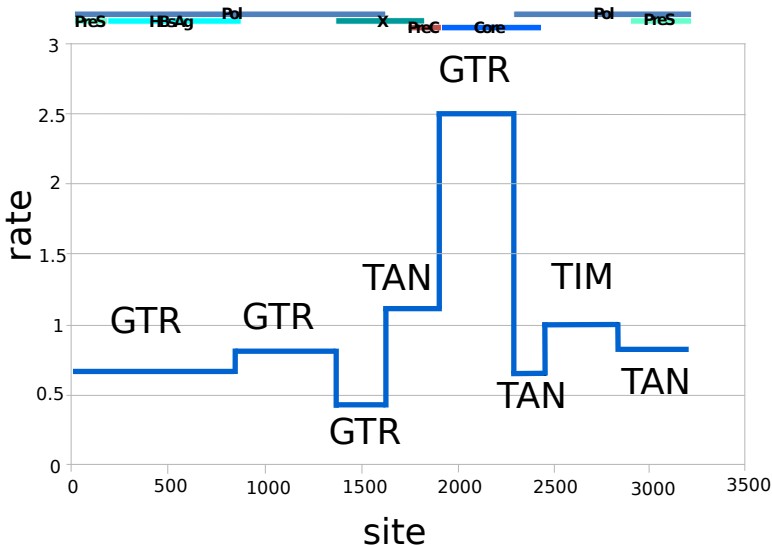

**Figure 1** **Relative rates (*y*-axis) and partitions among the HBV nucleotide site (*x*-axis), labelled with dominant substitution models for the partition following reversible-jump-based substitution model analysis.** The overlapping gene coding regions for the approximately 3,200 bp HBV genome are illustrated above the graph (Pol, polymerase). The genome was split into different partitions, since the HBV genome has many regions with overlapping coding frames, and each region can be expected to be governed by different evolutionary processes, explaining the difference in substitution models and average rates. GTR, General Time Reversible Model; TAN, Tamura and Nei; TIM, Transitional Model.

the only host for HBV in Arctic regions we can assume they share the same history. Archaeological evidence relevant to a rapid Inuit (Neo-Eskimo) migration to the Eastern Arctic from regions of Alaska starting approximately 900 to 700 YBP (sigma = 50 year; *Friesen & Arnold, 2008*; *McGhee, 2000*; *Raghavan et al., 2014*) was used to calibrate the root of genotype B coalescence, with an external calibrator estimate of 647 to 953 YBP. This migration displaced earlier Dorset Paleo-Inuit peoples with very little evidence for genetic or cultural interaction between the two populations (*Raghavan et al., 2014*), thus representing a re-setting of the human population distribution in the Eastern Arctic. Based on the mean external calibrator results, the HBV evolutionary clock rate for this study was estimated at $4.1 \times 10^{-5}$ substitutions per site per year (95% HPD interval [$3.1 \times 10^{-5}$, $5.1 \times 10^{-5}$]), which is in keeping with median rates from most literature sources, and takes into account the faster rate noted for sequences from HBeAg negative individuals (*Harrison et al., 2011*), of which most HBV/B5-infected individuals were in the present study due to a high prevalence of the precore stop codon mutation A1896 (Table 1; *Osiowy, Larke & Giles, 2011*). The estimated clock rate gave a root height for genotype B5 of 902 YBP (95% HPD Interval [803–1,001]) following DensiTree visualization of the posterior distribution over the set of trees (Fig. 2). This dating is consistent with minimum tMRCA estimates in the literature for genotype B5 (*Paraskevis et al., 2015*) and corresponds to the external calibrator employed. The DensiTree topology was found to be well resolved with approximately two-thirds of the clades having over 95% posterior support.

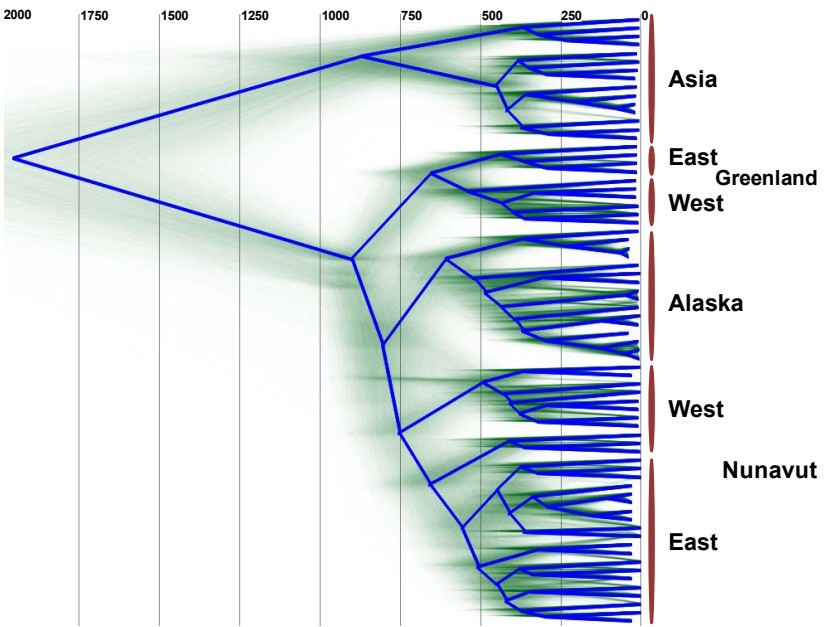

**Figure 2** **DensiTree showing clustering of HBV sequences within geographic locations.** The consensus tree is shown by the bold blue line and the estimated tMRCA years before present for tree nodes are shown at the top of the tree. Uncertainty of node heights and topology is shown by the transparent green lines. The clade on the top of the tree ("Asia") consists of HBV genotype B1 and B2 sequences, while all other sequences are HBV/B5. The clade B5 external calibrator estimate was calculated at 647 to 953 YBP based on archeological evidence of the Thule expansion at that time.

Unexpectedly, the estimated dates derived for the MRCA for HBV/B5 geographic lineages appear to shift in an east to west fashion, such that the tMRCA for the Greenlandic HBV/B5 node is approximately 650 YBP, while the average tMRCA for the Canadian and Alaskan HBV/B5 nodes is approximately 590 YBP. The earlier Greenlandic B5 branch is well supported by an 88.7% probability, while there is uncertainty as to whether subsequent branching is between Nunavut (east and west) and Alaska B5 taxa (47.2% probability) or between east Nunavut and Alaska/west Nunavut B5 (27.1% probability). This demonstrates that the root of HBV/B5 is associated with Greenlandic HBV genomic sequences, thus suggesting phylogenetic evolution of the virus during westward dispersal back into Alaska (Fig. 2).

## HBV/B5 population history and dispersal in the western circumpolar Arctic

Figure 3 shows the estimated effective number of HBV infections over time, where the *y*-axis represents the effective population size of HBV. A rapid expansion of the viral population is estimated to have occurred from approximately 400 YBP, which coincides with a rapid diversification shown in Fig. 2 within the same time frame. This coincides with archaeological evidence for a period of transformation in precontact Arctic society, during which a previously relatively homogeneous Thule society transformed into modern Inuit, starting approximately 500 YBP and continuing onwards for several centuries. This transformation is associated with population migrations, environmental shifts,

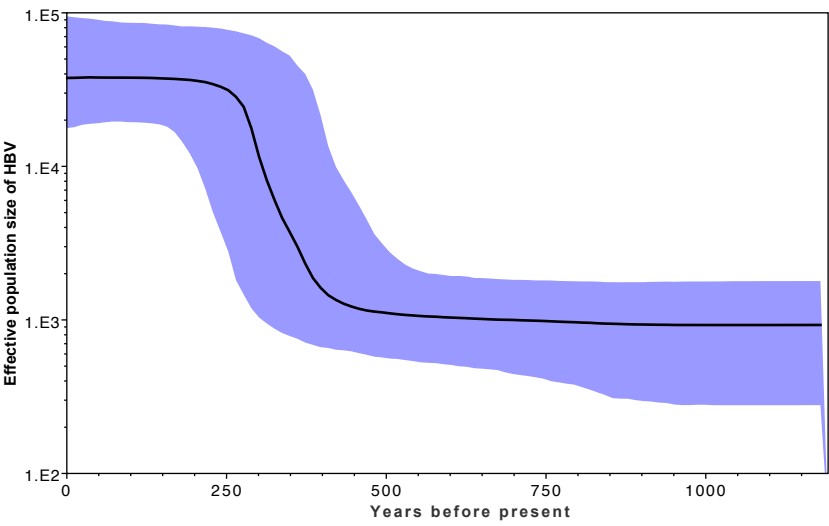

**Figure 3** **Effective population estimates of the HBV/B5 population (*y*-axis) over the years before present (*x*-axis) based on the Bayesian skyline plot.** The median effective population size is shown by the bold black line, with the 95% highest posterior density indicated in blue. The timeframe spans 1200 YBP to present day.

and contact with Europeans (*Finkelstein, Ross & Adams, 2009*; *Friesen, 2010*; *McGhee, 1994*). An estimate of the current HBV/B5 population size in the order of 100,000 is consistent with current estimated host population size estimates based on population statistics (*Central Intelligence Agency, 2015*; *Statistics Canada, 2014*; *United States Census Bureau, 2012*), in keeping with each infected person hosting an entire HBV quasi-species population (*Lauring & Andino, 2010*). If the molecular clock rate is increased, this results in reduced population size estimates; thus, confidence in the rate used is provided by the current estimates being consistent with current host populations (*Drummond & Bouckaert, 2015*). Coincident timing of the HBV population expansion with historical events provides further confidence in the clock rate employed.

The posterior distribution over the HBV/B5 tree sets is shown in Fig. 4 projected onto the map of the western circumpolar Arctic region as a function of increasing time. An animation of the landscape aware geographic model analysis which demonstrates the entire dispersal estimation over time and from which Fig. 4 is taken is provided as a Video S1. The blue line in Fig. 4 represents the path of the maximum clade confidence tree, shown in Fig. 2, and the set of trees representing the posterior is projected onto the map as transparent coloured dots indicating uncertainty in the path of dispersal, especially over northern Nunavut island regions. Thus, the model assumes that HBV/B5 follows a "random walk" through a geographic area, but it does not infer the nature of the host population; i.e., it is neither assumed that the walk occurs through empty space nor populated space, but the virus distribution may approximate the associated (uninfected) host population. The summary tree has good posterior support for most clades (about two third of the clades in the summary tree have over 95% posterior support), except those lower in the tree in Alaska and Nunavut, and there was a clear, monophyletic separation of geographic

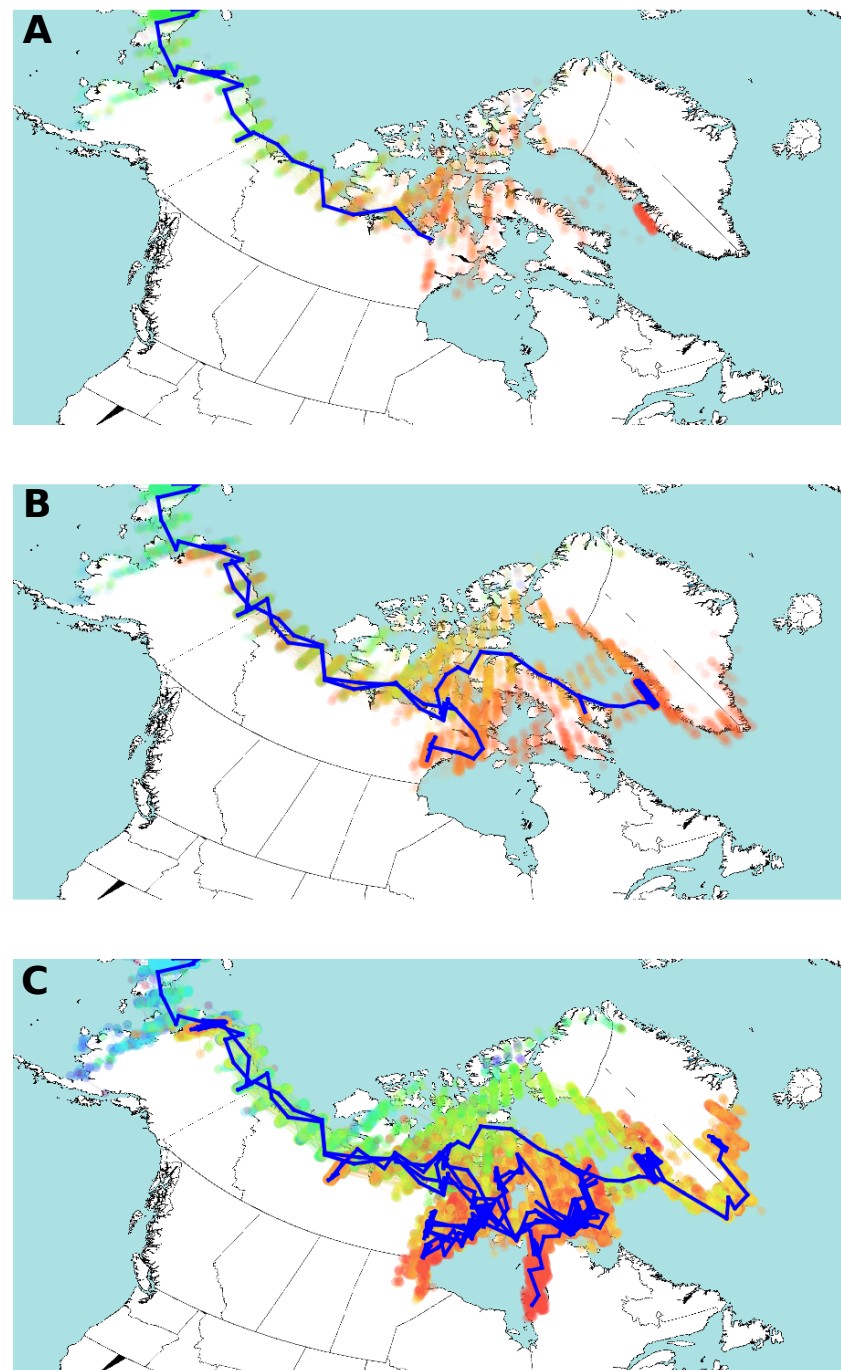

**Figure 4** **Estimated HBV dispersal routes into the high eastern Arctic, shown as a progression through time at times 900 YBP (A), 600 YBP (B) and present (C).** The blue line indicates the most plausible (highest probability) route of HBV dispersal based on landscape aware modelling. The coloured dots indicate the posterior distribution of dispersal. The dots are coloured by time, from light blue (earliest), green, yellow, orange, and red (most current). Background map from Wikimedia Commons (https://commons.wikimedia.org/wiki/File:World98.svg). An animation of the landscape aware geographic model analysis which demonstrates the entire dispersal estimation over time and from which Fig. 4 is taken is provided as Video S1.

HBV sequence clades, with 88.7% and 100% posterior support for the Alaska/Nunavut and Greenland/Nunavut/Alaska clades, respectively.

The advantage of the landscape aware model is that it allows features of the landscape, such as water, coastlines and inland areas, to be treated as having different dispersal rates. Original dispersal of HBV/B5 is estimated to have been more rapid along geographic coastal routes (*Helgason et al., 2006*; *Maxwell, 1985*; *Morrison, 1999*), similar to what has been described for HBV genotype C (*Littlejohn, Locarnini & Yuen, 2016*). The result of the landscape aware geographic model analysis suggests a rapid evolution of the HBV/B5 ancestor and spread along geographic coastal routes following its introduction into the Eastern Arctic from Alaska via Asia. As virus spread along the coast is estimated to be more rapid than inland, modelling tends to bend long branches along the coast, such as the branch from Asia into Alaska and further eastward. The modelling reconstruction shows Greenlandic viral evolution transpiring via the northern islands of Nunavut. The uncertainty of the actual dispersal into Greenland is shown by the maximum clade confidence tree vs. the posterior distribution (green dots in Fig. 4C) throughout Ellesmere Island. This uncertainty is likely due to the lack of HBV sequence coverage in the data set from the more northern regions of Nunavut. However, despite this limitation, the resulting analysis provides strong statistical support for an early dispersal route to Greenland through Ellesmere Island. Note the evident walk back towards Alaska in Figs. 4B and 4C.

## DISCUSSION

Our analysis of HBV/B5 sequences from the western circumpolar Arctic suggests that the virus evolved within Asia approximately 2000 YBP, with a later introduction into the Arctic. The eventual spread of the virus throughout the region possibly coincided with a sudden increase in the effective population of HBV, during the period of Thule Inuit transformation associated with expanded trade interactions and migrations of previously settled populations, creating a population of naïve hosts. Visualisation of phylogeographic analysis suggests that an initial rapid location of the virus to Greenland via Arctic coastal regions was followed by expansion and eventual HBV/B5 spread and evolution in a westward fashion, back to Alaska. A previous study investigating the origins of HBV also used a "coastal trail" model to explain the distribution of HBV genotype C throughout indigenous and relict populations of the Asia-Pacific region (*Littlejohn, Locarnini & Yuen, 2016*).

Established HBV subgenotypes have been located to East Asia at least 3000 YBP, based on phylogenetic analysis of HBV genotype C2 genomic sequences isolated from a 16th century Korean mummy (*Bar-Gal, 2012*). Furthermore, cladogenesis of the major HBV lineages and genotypes is estimated to have occurred approximately 20,000 YBP, due to major human population migrations (*Paraskevis et al., 2013*). The tMRCA for genotype B is estimated to have occurred anywhere from approximately 12 to 19.7 thousand YBP, depending on the clock model used (*Paraskevis et al., 2013*; *Paraskevis et al., 2015*). The use of external versus internal calibration approaches will influence phylodynamic reconstructions (*Zehender et al., 2014*). Thus, the use of a slower substitution rate ($5 \times 10^{-5}$ to $< 1 \times 10^{-6}$) together with a calibration based on ancient events, such as fossil or

human migration data, results in older ancestral nodes, which appears to support the hypothesis of HBV/human host co-expansion (*Paraskevis et al., 2015*) as a model of HBV origin. An external or remote calibration approach is appropriate for the present study as genotype B5 is uniquely isolated within western circumpolar Arctic indigenous populations and associations between HBV/B5 molecular and clinical outcome characteristics suggest a long-term history of the virus within this particular population (*Kowalec et al., 2013*; *Sakamoto et al., 2007*). As most HBV/B5 is associated with HBeAg negativity (*Osiowy, Larke & Giles, 2011*), the clock rate is expected to be somewhat faster than is observed with HBeAg positive individuals (*Harrison et al., 2011*).

The clock rate estimated in the present analysis results in the most recent common ancestor of HBV subgenotypes B1, B2 and B5 to be present approximately 2000 YBP. Assuming this rate is correct; the data demonstrates that B5 intra-evolution and differentiation likely began within HBV-infected populations resident in Asia. Thereafter, the genotype B5 coalescence among the three circumpolar Arctic regions in which HBV/B5 has been observed, occurs approximately 1,100 years later (i.e., approximately 900 YBP). Surprisingly, the data demonstrates that HBV/B5 from Greenland appears to be the "most ancient", implying that the variant present in modern Greenlandic Inuit diverged from the HBV/B5 ancestor over 650 YBP, while the variants that eventually form Canadian and Alaskan HBV/B5 monophyletic clades started to diverge shortly thereafter. *Paraskevis et al. (2013)* observed a similar pattern of HBV/B5 viral evolution, such that the HBV/B5 median tMRCA estimate from Alaska was determined to be more recent than that from Canada or Greenland.

Based on archaeological evidence, the eastern Arctic was inhabited by the Paleo-Inuit Late Dorset peoples from approximately 700 to 1300 YBP (*Friesen, 2004*; *Maxwell, 1985*). Prior to this period, the chronology of various cultural groups within the Arctic stems from a migration of earlier Paleo-Inuit peoples from Siberia over 4000 YBP (*Achilli et al., 2013*; *Friesen, 2016*; *Gilbert et al., 2008*). Archaeological evidence further shows that Neo-Eskimo people, also genetically and culturally associated with ancient Siberian peoples (*Raghavan et al., 2014*), rapidly migrated from regions of Alaska through to the eastern Arctic starting approximately 900 to 700 YBP (*Friesen & Arnold, 2008*; *McGhee, 2000*; *Moltke et al., 2015*; *Morrison, 1999*). The earliest Neo-Eskimo migrants to the Eastern Arctic are known as Thule, and are accepted as the ancestors of modern Inuit, with little evidence of genetic mixing between Late Dorset and Thule peoples (*Raghavan et al., 2014*). By 600–500 YBP, transformation of the homogeneous Thule culture into more diverse cultural groupings had started in a process that lasted for several centuries, based on archaeological observation of shifts in regional populations, settlement patterns and social organization (*Friesen, 2010*). The impetus for this transformation has been speculated to involve climatic changes leading to increased sea ice and alteration in subsistence sources (*Finkelstein, Ross & Adams, 2009*) as well as changing internal social networks and increasing external contact and trade with European explorers, whalers and merchants (*Friesen, 2010*; *McGhee, 1994*).

The genotype B5 genomic coalescence determined in the present study suggests that HBV/B5 was present in the Neo-Eskimo populations that travelled eastward and populated northwestern Greenlandic sites. Recent evidence points to the pioneering Greenlandic

Inuit population as a historically isolated founder population, demonstrated by decreased nucleotide diversity (*Moltke et al., 2015*). Prolonged isolation among a relatively small, HBV-endemically infected population, may result in viral adaptation, which in turn is associated with a slower viral evolutionary rate (*Lin et al., 2015*; *Zehender et al., 2014*), possibly resulting in slow or static nucleotide divergence up until the time of HBV population expansion. The Bayesian skyline plot data shows HBV/B5 expansion between 400 to 250 YBP, coincident with timing of a complex transformation of the Neo-Eskimo Thule people into more diverse cultural forms consistent with modern Inuit. Counter-intuitively, the evolutionary history of HBV/B5 within the Arctic appears to start with a Greenlandic ancestor, by way of an earlier Asian ancestor. However, this pattern actually fits one of the central aspects of current models of the Thule migration from Alaska to the eastern Arctic. Archaeological site distributions indicate that the initial migration was rapid, and involved groups from more than one area in the Alaska/Bering Strait region. The earliest populations in the northwest Greenland/northern Ellesmere Island region are known as "Ruin Island Thule", and are widely accepted as; (a) among the earliest, or possibly the absolute earliest, Thule population in the eastern Arctic; (b) somewhat different from early Thule elsewhere in the Canadian Arctic, based on their material culture; and (c) originating in the Bering Strait region, rather than northern Alaska where other migrating Thule populations are likely to have come from (*Friesen, 2016*; *Gulløv & McGhee, 2006*; *Marchani, Rogers & O'Rourke, 2007*; *McCullough, 1989*; *Morrison, 1999*). The HBV/B5 pattern is consistent with this reconstruction, and could represent an initial Ruin Island Thule population carrying B5 from the Bering Strait region into northwest Greenland, followed by migration of other Alaskan Thule populations into the Canadian Arctic, followed in turn by the spread of B5 back through those later Thule populations from east to west. This east to west 'back-migration' within the high Arctic has been suggested through tMRCA dating of the Y chromosome (*Olofsson et al., 2015*), mtDNA variant analysis (*Raff et al., 2015*; *Tamm et al., 2007*), SNP genotyping (*Reich et al., 2012*) and linguistic analysis (*Hammarström et al., 2015*; *Raff et al., 2015*; *Sicoli & Holton, 2014*).

Once large scale B5 expansion occurred, horizontal transmission and viral "colonization" likely resulted in a more rapid substitution rate (*Lin et al., 2015*), eventually leading to an HBV/B5 variant having characteristics observed in modern populations, such as a significantly higher nucleotide diversity (*Kowalec et al., 2013*; *Osiowy et al., 2006*) compared to other HBV genotypes which infect indigenous populations, and a high prevalence of the precore A1896 stop codon mutation. HBV/B5 has been hypothesized to function as an "adaptor" variant, such that it excels at escaping host immune selection through its rapid mutation rate, but likely at the cost of high replicative activity, or viral load, thus resulting in lowered transmission. A continued association with the newly infected, yet regionally isolated transitional populations throughout the Arctic would further allow for adaptation in the form of pathogen attenuation, permitting a balance between viral escape leading to persistence, and host immune control, resulting in a lack of immune-mediated liver disease (*Minuk et al., 2013*).

The present study has several limitations, including sampling bias and a paucity of HBV/B5 sequence data from certain regions. The theory of HBV/B5 evolution and spread

in an eastward direction would possibly be supported by obtaining HBV/B5 samples from regions of Siberia to be included in phylogeographical analysis; however, such samples have not yet been identified. HBV co-infection with hepatitis D virus (HDV) has been shown to be high in certain West Greenland communities (*Børresen et al., 2010*; *Langer, Frösner & Von Brunn, 1997*) and it is possible that co-infection may influence the short-term evolutionary rate of HBV in these individuals, thus affecting the molecular clock rate. Although HBV-infected persons from whom B5 sequences were obtained from East Greenland were found to be negative for antibody to HDV, the same is not known for the West Greenland sequences.

## CONCLUSIONS

Through our novel phylogeographic approach, the origin and spread of HBV/B5 throughout the western circumpolar Arctic has been estimated to occur coincident with the movement and settlement of Neo-Eskimo populations within the past 1000 years. Results of this study and the knowledge of the unique association of HBV/B5 with indigenous populations of this region, including historical endemicity and a benign clinical outcome, further support the hypothesis of long-term co-evolution between virus and host, and illustrate the many intricate interactions between a specific variant and an infected population over time.

## ACKNOWLEDGEMENTS

The authors gratefully acknowledge Elizabeth Giles and Chris Huynh for excellent technical assistance in HBV sample sequencing and thank all study participants.

### Funding
The authors received no funding for this work.

### Competing Interests
The authors declare there are no competing interests.

### Author Contributions
- Remco Bouckaert conceived and designed the experiments, performed the experiments, analyzed the data, wrote the paper, prepared figures and/or tables, reviewed drafts of the paper.
- Brenna C. Simons and Henrik Krarup contributed reagents/materials/analysis tools, reviewed drafts of the paper.
- T. Max Friesen analyzed the data, wrote the paper, reviewed drafts of the paper.
- Carla Osiowy conceived and designed the experiments, performed the experiments, analyzed the data, contributed reagents/materials/analysis tools, wrote the paper, prepared figures and/or tables, reviewed drafts of the paper.

## Human Ethics

The following information was supplied relating to ethical approvals (i.e., approving body and any reference numbers):

The Health Canada/Public Health Agency of Canada research ethics board, the University of Manitoba research ethics board, the Alaska Area (Indian Health Service) and the Centers for Disease Control and Prevention institutional review boards, and the Commission for Scientific Research in Greenland all provided ethical approval to carry out past studies from which specimens were approved for HBV DNA detection and sequence analysis.

## DNA Deposition

The following information was supplied regarding the deposition of DNA sequences:

The new sequences generated for this study were deposited to GenBank under accession numbers KP659219–KP659228 and KP659230–KP659255.

## Data Availability

The XML file including all data for BEAST 2 analysis is available as a Supplementary File. An animation of the landscape aware geographic model analysis which demonstrates the entire dispersal estimation over time, and from which Fig. 4 is taken, is available as a Supplementary File.

## Supplemental Information

Supplemental information for this article can be found online at http://dx.doi.org/10.7717/peerj.3757#supplemental-information.

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
