# Peer review of "Tracing hepatitis B virus (HBV) genotype B5 (formerly B6) evolutionary history in the circumpolar Arctic through phylogeographic modelling"

_PeerJ, doi:10.7717/peerj.3757_

## Round 0.1 · original submission · Minor Revisions

While brief, the reviews are in general agreement that only minor revisions are required. Please address the concerns they have raised.

Reviewer 1 ·

Basic reporting

The authors explored the evolutionary spread of HBV/B5 in the circumpolar arctic, using phylogeographic model, to support the hypotheisis of HBV/host coexpansion. Overall the paper is nicely written, and supply useful information for illustrating the host-pathogen adaptation.

Experimental design

Data and methods are clearly described.

One minor questions: Why HBV/B1 and B2 are included in the phylogenetic analysis even they have not been observed in circumpolar arctic area?

Validity of the findings

The results and discussions are well stated.

minor question: For fig 4, why use 900, 600 YBP and present? As from the results and discussion, 400-500 YBP is shown as rapid diversification, and also is the population migration transformation period.

Reviewer 2 ·

Basic reporting

Overall, the manuscript is clearly written in professional, unambiguous language. However, there is 1 long sentence that needs to be further clarified for easier understanding: page 7, line 11 ~ 13.
Background information is clearly introduced to show context, and the literatures are highly relevant and well referenced. Please also add references for the 6 substitution models used on page 7 line 4.
Article structure is professional. Figures are relevant, high quality and well labeled & described.
Sequence data and BEAST2 parameter settings in xml format are also provided.

Experimental design

The research topic is within Scope of PeerJ and should be interesting to the readership of the journal.
The research question is well defined – investigate the origin, evolution and spread of HBV/B5. I suggest the authors provide stronger justification for their study, specifically, please elaborate on the knowledge gap being filled.
The research is well designed and performed to a high technical & ethical standard. Methods are described with sufficient detail to replicate.

Validity of the findings

Data is robust and statistically sound. Conclusions are well stated and linked to the original research question & limited to supporting results. A relatively long Discussion section is written and it provides (1) justification of the validity of the parameters the authors chose, (2) speculation how their findings supporting host-pathogen balance and (3) brief discussion about the limitation of their research. While all these are informative materials, I suggest the authors further distill the key points and write in a more concise manner for easier understanding. Note this modification isn't essential, the authors can decide by themselves.

---

## Round 0.2 · accepted · Accept

All the concerns have been properly addressed.

Reviewer 1 ·

Basic reporting

no comment

Experimental design

no comment

Validity of the findings

no comment

Additional comments

The authors have responded to all my questions clearly. I have no further comments/suggestions.

Reviewer 2 ·

Basic reporting

No comment.

Experimental design

No comment.

Validity of the findings

No comment.

Additional comments

All my comments were considered by the authors and the modifications are acceptable. I have no further suggestions.